# Prevalence of Licit and Illicit Drugs Use during Pregnancy in Mexican Women

**DOI:** 10.3390/ph15030382

**Published:** 2022-03-21

**Authors:** Larissa-María Gómez-Ruiz, Emilia Marchei, Maria Concetta Rotolo, Pietro Brunetti, Giulio Mannocchi, Aracely Acosta-López, Ruth-Yesica Ramos-Gutiérrez, Mary-Buhya Varela-Busaka, Simona Pichini, Oscar Garcia-Algar

**Affiliations:** 1Servicio de Neonatología, División de Pediatría, Nuevo Hospital Civil de Guadalajara “Dr. Juan I. Menchaca”, Guadalajara 44600, Mexico; lmgomez@hcg.gob.mx (L.-M.G.-R.); ara.acosta@gmail.com (A.A.-L.); dr_yesi0612@yahoo.com.mx (R.-Y.R.-G.); mbvarela@hcg.gob.mx (M.-B.V.-B.); 2Departamento de Cirugía y Especialidades Médico-Quirúrgicas, Universidad de Barcelona, 08036 Barcelona, Spain; ogarciaa@clinic.cat; 3National Centre on Addiction and Doping, Italian National Institute of Health (ISS), V.Le Regina Elena 299, 00161 Rome, Italy; emilia.marchei@iss.it (E.M.); mariaconcetta.rotolo@iss.it (M.C.R.); 4Unit of Forensic Toxicology, Section of Legal Medicine, Department of Excellence of Biomedical Scienc-Es and Public Health, Marche Polytechnic University, 60126 Ancona, Italy; pietrobrunetti40@gmail.com; 5School of Law, University of Camerino, 62032 Macerata, Italy; giulio.mannocchi@unicam.it; 6Neonatology Unit, Hospital Clinic-Maternitat, ICGON, BCNatal, C/Sabino Arana 1, 08028 Barcelona, Spain

**Keywords:** pregnancy, illicit drugs, tobacco, prenatal exposure, maternal hair

## Abstract

For the first time, the present study employed hair testing to investigate the prevalence of classical drugs of abuse and new psychoactive substances use during gestation in a cohort of 300 Mexican pregnant women. An interview was conducted to collect data on sociodemographic aspects of the patients, and a 9 cm-long hair strand was taken from the back of the head of each mother one month after delivery. A validated ultra-high-performance liquid chromatography–high-resolution mass spectrometry method was used for the screening of classic drugs, new psychoactive substances, and medications in maternal hair. Out of 300 examined hair samples from pregnant women, 127 (42.3%) resulted positive for psychoactive substances: 45 (35.4%) for cannabis only, 24 (18.9%) for methamphetamine only, 13 (10.2%) for cocaine only, 1 (0.3%) for heroin, 1 for N-N-dimethyltryptamine (0.3%), 1 for ketamine (0.8%), and 35 (16.3%) for more than one psychoactive substance. Furthermore, seven samples (2.3%) resulted positive for new psychoactive substances (NPS): two samples for synthetic cannabinoids, two for synthetic cathinones, and three for nor-fentanyl, and 3.3% of women hair resulted positive for anticonvulsant, antidepressant, and antipsychotic medications. Finally, 83 women hair samples (27.7%) tested positive for nicotine. Nonsteroidal anti-inflammatory drugs (NSAIDs) and other painkillers (60.0%), medications for the treatment of nausea and vomiting (12.3%), antihistamines (8.7%) and nasal/sinus decongestants (6.7%), cough suppressants (5.0%), and bronchodilator agents (5.0%) were also detected in pregnant women hair. The gestational use of psychoactive substances and exposure to tobacco smoke, assessed by hair testing, were associated with a significantly younger age and with a low education grade of the mothers (*p* < 0.005). This study provides a significant preliminary indication of the under-reported gestational consumption of licit and illicit psychoactive and pharmacologically active drugs in a Mexican environment, showing the value of toxicological and forensic analyses in the global effort to determine the health risks caused by classic drugs and new psychoactive substances during pregnancy.

## 1. Introduction

The period of pregnancy is particularly delicate and requires special attention to protect the health of both the woman and her child. During pregnancy, the use of substances, whether legal or illicit, may cause health, social, and legal negative consequences [1,2].

Women might be hesitant to seek treatment, because an intense stigma exists against substance abuse, and licit and illicit substance use prevalence during pregnancy remains underestimated [3,4,5].

While information on the use of legal and illicit substances in pregnancy is available for North America and Europe, it is worth noting the relative paucity of studies and research conducted in Latin America, in particular, in Mexico. In this regard, it has to be considered that drug use during pregnancy is a demographic problem that is becoming increasingly important as it affects health, education, the economy, and social and cultural relationships [6,7,8,9].

Information about the use of tobacco and illicit drugs as well as of prescription drugs during pregnancy in Mexico is limited [9,10,11,12,13]. Moreover, generally speaking, the real prevalence of drugs consumption among pregnant women is difficult to ascertain, as studies on this are mainly based on data collected through interviews, which, although anonymous, tend to underestimate the problem due to the fear of stigmatization of the participants [8,14,15,16,17]. Indeed, the average prevalence of illicit drug use in pregnant women determined by studies based on questionnaires or interviews resulted to be less than 2% (1.65%), whereas in studies based on toxicological analyses, it was more than 10% (12.28%, i.e., 7.4 times higher) [1].

Therefore, pharmacotoxicological laboratories, through the measurement of biomarkers of substances exposure in maternal and neonatal biological matrices, play an essential role in assessing the real maternal prevalence of illicit substance use and consequent prenatal exposure to them [2,18,19,20,21,22,23], allowing the generation of appropriate policies and interventions that are supportive, non-judgmental, and empathetic and help women to withdraw from drugs [17,24].

Among the biological matrices investigated to evaluate gestational licit and illicit drugs consumption, maternal hair resulted to be the most suitable for covering the entire pregnancy period. Indeed, considering a mean hair growth of 1 cm per month, a 9 cm hair shaft, cut one month after delivery, corresponds to the whole 9 months of pregnancy [21,25,26].

Recently, we developed an ultra-high-performance–high-resolution mass spectrometry assay to screen several xenobiotics in hair samples [27]. The present study employed this validated method to investigate, for the first time, the consumption of classical drugs of abuse, new psychoactive substances, and prescription medicines in a cohort of 300 Mexican pregnant women.

## 2. Results

### 2.1. Participants’ Characteristics

During the 3-month recruitment period, 300 women received delivery care at Nuevo Hospital Civil de Guadalajara “Dr. Juan I. Menchaca”, Guadalajara, Jalisco, México, and met the eligibility criteria to participate in the study. Table 1 shows the sociodemographic profiles of the women enrolled in the study. Only 8 women (2.7%) disclosed illicit drugs use during pregnancy, 17 (5.7%) and 37 (12.3%) women declared consumption of tobacco and alcohol, respectively, and 3 women (1.0%) declared the use of inhalants (with no other specification).

### 2.2. Licit and Illicit Psychoactive Substances Assessment in Maternal Hair

With a total of 300 examined pregnant women, hair samples from 127 (42.3%) women resulted positive for psychoactive substances: 45 (15%) for cannabis only, 24 (8%) for methamphetamine only, 13 (4.3%) for cocaine only, 1 (0.3%) for heroin only, 1 (0.3%) for N-N-dimethyltryptamine only, 1 (0.3%) for ketamine only, and 35 (11.6%) for more than one psychoactive substance (Table 2 and Table 3).

Furthermore, seven samples (2.3%) resulted positive for new psychoactive substances (NPS): two samples for synthetic cannabinoids, two for synthetic cathinones and three for nor-fentanyl. Finally, 3.3% of the examined samples resulted positive for anticonvulsants (three for carbamazepine), antidepressants (three for citalopram and three for sertraline), and antipsychotics (one case positive for haloperidol) (Table 2).

Self-reported illicit drug use (0.0% vs. 6.3%, chi-square *p* < 0.05) and exposure to tobacco smoking (1.7% vs. 10.9%, chi-square *p* < 0.05) were significantly associated with illicit drugs consumption (Table 4).

Illicit drugs consumption was associated with a significantly younger age of the mothers (22.6 ± 5.6 vs. 25.2 ± 6.5, *t*-Student *p* < 0.05). Moreover, users of more than one drug were significantly younger than users of only cocaine (21.5 ± 4.8 vs. 25.7 ± 7.1, *t*-Student *p* < 0.05) (Table 4).

The academic profiles were different between cannabis, polydrug, and NPS consumers. Education to secondary level was more frequent in NPS users than in cannabis users (85.7 vs. 44.4, chi-square *p* < 0.05) and in those consuming more than one drug (85.7 vs. 34.3, chi-square *p* < 0.05) (Table 4).

### 2.3. Passive and Active Smoking Assessment in Maternal Hair

Maternal hair samples were negative for nicotine, used as a biomarker of tobacco smoke, in 217 (72.3%) cases and positive in 83 (27.7%) cases. Specifically, 68 samples (22.7%) showed a concentration of nicotine <3 ng/mg, indicating low exposure to environmental tobacco smoke (ETS), 14 samples (4.7%) had a concentration between 3 and 18 ng/mg, indicating medium exposure to ETS, and only one sample (0.3%) contained >18 ng/mg of nicotine, indicating active smoking or high exposure.

Self-reported tobacco smoke (2.8% vs. 13.3%, chi-square *p* < 0.05) and exposure to ETS (37.2% vs. 57.8%, chi-square *p* < 0.05) were significantly associated with hair positivity for nicotine. In addition, positive cases were significantly associated with self-reported exposure to familiar ETS (61.4% vs. 33.2%, chi-square *p* < 0.05) (Table 5).

Tobacco use was associated with a significantly younger age (23.0 ± 5.2 vs. 24.6 ± 6.6, t-Student *p* < 0.05) and with a low education grade. No difference was found in social status, while when considering the civil status, married women tended to smoke less (21.7% vs. 8.4, chi-square *p* < 0.05) than cohabiting women (60.8% vs. 75.9%, chi-square *p* < 0.05).

### 2.4. Prescription Drugs Assessment in Maternal Hair

The principal detected prescription drugs were nonsteroidal anti-inflammatory drugs (NSAIDs) and pain killers (60.0% hair samples) and, among these, paracetamol (42.0%), alone and in combination with other NSAIDs and/or pain killers, was the most used prescription drug, followed by metamizole (32.7%, alone and in combination with other NSAIDs and/or pain killers). Phenazone (8.7%) and nimesulide (3.0%) were also detected. Other detected prescription drugs were those used against nausea and vomiting (e.g., metoclopramide and ondansetron) (12.3%), antihistaminic drugs (e.g., doxylamine, chlorpheniramine, loratadine) (8.7%), cough suppressants (e.g., dextromethorphan, dropropizine) (5%), and nasal/sinus decongestants (e.g., pseudoephedrine) (6.7%).

### 2.5. Neonatal Profiles Associated with Gestational Consumption of Psychoactive Drugs and Exposure to Tobacco Smoke

Table 6 shows the neonatal characteristics at birth in relation to the maternal consumption of psychoactive drugs assessed by hair analysis.

Generally speaking, no significant differences were observed between newborns not exposed and exposed to maternal consumption of psychoactive drugs or to tobacco smoke. However, when considering specifically the single psychoactive substances, infants prenatally exposed to cannabis only were significantly heavier than all the others (3187.8 ± 633.5 vs. 3004.6 ± 560.7, *t*-Student *p* < 0.05) and than newborns not exposed to any illicit drugs (3187.8 ± 633.5 vs. 2969.1 ± 582.8, t-Student *p* < 0.05).

## 3. Discussion

Even if the effects of gestational consumption of psychoactive drugs on fetal development and pregnancy complications are widely known, the prevalence of licit and illicit psychoactive compounds consumption during pregnancy seems not to decrease in developed societies [1,2,5,6,7]. In this study, we demonstrated that also in a developing country, such as Mexico, the gestational consumption of licit and illicit psychoactive substances is not a negligible phenomenon. In addition, the results of this study confirmed that, as in previous studies, maternal interviews underestimate consumption, which can be objectively assessed by hair analysis [15,17,24].

Cannabis was the psychotropic drug mainly abused by this cohort of pregnant women, showing a prevalence of 15.0%, which rose to 23.0% when including cases positives for more than one drug. The other two most consumed illicit drug were methamphetamine, with a prevalence of 8.0% increasing to 17.0% when considering cases positives for more than one drug, and cocaine, with a 4.3% positivity rising to 9.0% when including cases positives for more than one drug.

These results are in agreement with the prevalence of drugs of abuse consumption reported in the general Mexican population [28,29,30], which provides high reliability to our data. Indeed, since the 1990s, there has been a significant increase in the illicit use of drugs in Mexico, with cannabis and cocaine being the substances most often used, followed by amphetamine-type drugs (the most common being methylenedioxymethamphetamine). Actually, methamphetamine use has also become a major public health concern in Mexico as a consequence of the country’s growing role as a major producer of this substance [28,29,31].

Data from the most recent Mexican report on the consumption of psychoactive drugs [30] confirm that the substances whose consumption causes the greatest demand for treatment are amphetamine-type stimulants, such as amphetamine, methamphetamines, ecstasy, or stimulants for medical use, with a total of 30.2% of the total cases, followed by marijuana, with 15.1% of the total cases.

The report also highlighted a significant increase in cases of fentanyl use, and even if several Latin countries, including Mexico, have reported the identification of NPS such as synthetic cathinones and synthetic cannabinoids in patients with local seizures, information on their prevalence of use has not been available up to now [31]. Our results showed a prevalence of fentanyl and other NPS of 2.3%, which rose to 3.7% when comprising cases positives for more than one drug. It has to be said that, in this study, it was decided to consider fentanyl as an NPS and, more specifically, as a new synthetic opioid, whose recreational use in place of heroin has recently been exponentially increasing [31].

Although Mexico is rich in psychoactive plants (Peyote, Psilocybin mushrooms, *Salvia divinorum*, *Psychotria viridis*, etc.) only one sample was found positive for the psychedelic N,N-dimethyltryptamine.

The concentration of the detected psychoactive drugs would have been important to differentiate between sporadic and chronic consumption during gestation. Unfortunately, our developed method focused only on the identification of the highest number of pharmacologically active substances, even in the absence of pure standards, and quantitative analyses were not performed. In any case, the mere presence of a drug in the hair of pregnant women is an index of consumption, and these drugs are in any case harmful and prohibited during the development of the fetus. Another limitation of the present study was that the developed method showed low sensitivity in the measurement of the gestational alcohol consumption biomarker ethylglucuronide in hair [27]. As reported, a specific sample extraction coupled with an exclusive UHPLC–HRMS method for EtG is currently under evaluation.

The only performed quantification was that of nicotine, since it is specifically required to identify gestational active and passive smokers.

In this regard, we found only one case of active smoking or eventually high exposure to ETS, but 27.4% of pregnant women were exposed to ETS, though only 5.7% women declared exposure. Our results highlight how parental smoking is a significant determinant of the risk of fetal exposure to nicotine during pregnancy and, as such, is a major and entirely avoidable health risk for both women and their child. Indeed, pregnant women should be protected from exposure to smoking, especially by family members.

Maternal use of prescription drugs during pregnancy is common and concerns medications to treat allergies and respiratory and gastrointestinal conditions, in addition to general analgesics [2,32,33,34]. In this study, 82.6% of pregnant women used a prescription drug, with paracetamol being the most used and generally considered safe during pregnancy [33,34]. A careful risk/benefit assessment for each drug should be done before prescribing them in pregnancy, especially when considering psychoactive substances such as the ones identified in this study, i.e., anticonvulsants, antidepressants, and antipsychotics, which can be misused, with unknown consequences for the fetus.

Notwithstanding the evidenced gestational consumption of psychoactive drugs in this examined cohort, no significant correlations were found between neonatal outcomes, mothers’ hair results, and interview answers. This is in contrast with previous studies showing a correlation between gestational consumption of tobacco and drugs of abuse and increased risk of spontaneous abortions, reduction in neonatal birth weight and birth length, newborn head circumference, and signs of a more severe neonatal abstinence syndrome [27,35,36,37,38]. It can be hypothesized that in this examined cohort, psychoactive drugs consumption during pregnancy was sporadic and, even if identifiable by hair testing, did not affect offspring features.

## 4. Materials and Methods

### 4.1. Participants

The study was a prospective observational study carried out in Nuevo Hospital Civil de Guadalajara “Dr. Juan I. Menchaca”, Guadalajara, Jalisco, México, between 1 November 2019 and 31 January, 2020. The cohort consisted of pregnant women who received delivery care at the Nuevo Hospital Civil de Guadalajara “Dr. Juan I. Menchaca”, Guadalajara, Jalisco, México, and accepted to participate in the study, signing the informed consent.

Three–four weeks after delivery, a medical doctor interviewed the women, and hair samples were collected. The interview was performed with a standardized survey, which included questions about maternal sociodemographic characteristics (age, nationality, profession, studies, marital status, etc.) and maternal drug habits before and during pregnancy. The survey data and collected hair were coded in order to secure the participants’ privacy, and the local Human Research Ethics Committee (CONBIOETICA-14-CEI-008-20161212) approved the study protocol.

Moreover, the following neonatal outcome measures were recorded: estimated gestational age (EGA) at delivery (weeks), birth weight (grams), head circumference (centimeters), length (centimeters), and pathologies.

### 4.2. Hair Analysis

Hair samples (at least 9 cm, so to take into account the entire gestational period) were analyzed by a validated previously published assay employing an overnight incubation at 37 °C in 0.5 mL of a mixture of 2 mM ammonium formate, methanol, and acetonitrile (50/25/25, *v*/*v*/*v*), followed by liquid chromatography–high-resolution mass spectrometry analysis for a target screening of more than 1000 substances including illicit drugs, new psychoactive drugs, and prescription drugs [27]. Since the proposed methodology was only intended for the qualitative screening of hair samples, the limits of detection and of identification were estimated as previously reported [27]. Hair samples were considered positive when substances were found above their limit of identifications, ranging from 0.02 to 0.10 ng/mg hair.

### 4.3. Data Analysis

The data obtained from mothers’ hair biomarkers testing and interviews were recorded in a Microsoft Office Excel 10 spreadsheet. To obtain the demographic profiles associated with the use during pregnancy of licit and illicit psychoactive substances, the group with negative toxicological screening results for all tested substances was compared with the group that tested positive for any of licit and illicit substance, the groups positive for only for one psychoactive substance, and the group positive for more than one psychoactive substance. Values are expressed as the mean standard deviation or frequency (percentage).

Associations between sociodemographic and lifestyle characteristics of the pregnant women with hair biomarkers for psychoactive substance use and smoking behavior were performed by an independent *t*-test for quantitative variables and a chi-square test for qualitative variables. Statistical significance was set at *p* < 0.05.

## 5. Conclusions

For the first time in a Mexican cohort, this study objectively assessed licit and illicit drug consumption in a pregnant women population. Our results confirm the usefulness of maternal hair analysis to evidence drug use during pregnancy and how the measurement of exposure biomarkers in this matrix is essential to demonstrate real consumption.

The early detection of licit and illicit drug use during pregnancy using toxicological analyses in biological matrices and/or effective screening programs combined with the participation of qualified professionals can help to define the most appropriate measures to avoid the gestational consumption of psychoactive drugs and consequent prenatal exposure.

## Figures and Tables

**Table 1 pharmaceuticals-15-00382-t001:** Sociodemographic profiles of the women enrolled in the study.

Variables (%)	All Cases (*n* = 300)
Nationality	
Mexican	98.7
Others	1.0
NA	0.3
	Age (mean ± SD)
*Academic level*	
No study	1.3
Primary school	24.7
Secondary school	47.7
High school	21.3
College	5.0
*Social class*	
Housewife	81.3
Student	2.0
Employed	7.7
Day worker	0.7
Dealer	4.7
NA	3.6
*Civil status*	
Single	16.0
Married	18.0
Cohabitant	65.0
Widow	1.0
*Previous pregnancy*	
0	32.7
1	25.0
2	18.7
>2	23.6
*Self-reported use*	
Illicit drugs	2.7
Tobacco	5.7
Alcohol	12.3
Inhalants	1.0
*Offspring’s pathologies*	
YES	16.0

NA: not available.

**Table 2 pharmaceuticals-15-00382-t002:** Percentage of licit and illicit substances in hair samples from all the enrolled women (*n* = 300) and in the 127 women whose hair was positive for any psychoactive drug.

	All Cases (*n* = 300)	Maternal Hair Positive for Licit and Illicit Drugs (*n* = 127)
Classic drugs of abuse (*n* = 120)		
Cannabis (*n* = 45)	15.0%	35.4%
Methamphetamine (*n* = 24)	8.0%	18.9%
Cocaine (*n* = 13)	4.3%	10.2%
More than one drugs (*n* = 35)	11.6%	27.5%
Heroin (as 6-Monoacetylmorphine) (*n* = 1)	0.3%	0.8%
N-N-dimethyltryptamine (*n* = 1)	0.3%	0.8%
Ketamine (*n* = 1)	0.3%	0.8%
New Psychoactive substances (*n* = 7)		
Synthetic cannabinoids (*n* = 2)	0.6%	1.6%
Synthetic cathinones (*n* = 2)	0.6%	1.6%
Fentanyl (*n* = 3)	1.0%	2.4%
Prescription psychoactive drugs (10)		
Anticonvulsants (*n* = 3)	1.0%	0.8%
Antidepressants (*n* = 6)	2.0%	2.4%
Antipsychotics (*n* = 1)	0.3%	0.8%

**Table 3 pharmaceuticals-15-00382-t003:** Psychoactive substances found in hair from polyconsumer pregnant women.

Case Number (*n* = 35)	Substances Detected
13	Methamphetamine THC
9	Cocaine THC
2	Cocaine Methamphetamine, amphetamine, and ethylamphetamine THC
3	Methamphetamine, amphetamine
1	Methamphetamine, amphetamine, and ethylamphetamine
1	Methamphetamine Methcathinone (Ephedrone)
1	Methamphetamine Ketamine, Norketamine
1	Cocaine Methamphetamine, amphetamine
1	4-Fluoroamphetamine (4-FA) 4-methylethcathinone (4-MEC) Methadone, EDDP
1	Cocaine Methamphetamine
1	4-fluoroamphetamine (4-FA) Methadone, EDDP
1	EDDP, Methadone Fentanyl Nor-fentanyl

**Table 4 pharmaceuticals-15-00382-t004:** Sociodemographic profiles associated with gestational consumption of illicit drugs assessed by hair testing.

					Maternal Hair					
Variables (%)	Negative for Illicit Drugs (*n* = 173)	Positive for Illicit Drugs (*n* = 127)	Positive for Cannabis (*n* = 45)	Positive for Cocaine (*n* = 13)	Positive for Methamphetamine (*n* = 24)	Positive for More than One Drug (*n* = 35)	Positive for NPSs (*n* = 7)	Positive for 6-MAM (*n* = 1)	Positive for N,N-DMT (*n* = 1)	Positive for Ketamine (*n* = 1)
*Nationality*										
* Mexican*	98.8	98.4	97.8	92.3	100	100	100	100	100	100
* Others*	0.6	1.6	2.2	7.7	0.0	0.0	0.0	0.0	0.0	0.0
NA	0.6	0.0	0.0	0.0	0.0	0.0	0.0	0.0	0.0	0.0
Age (mean ± SD)	25.2 ± 6.5	22.6 ± 5.6 *	23.4 ± 5.4	25.7 ± 7.1	21.8 ± 6.0	21.5 ± 4.8 *	20.6 ± 5.0	27.0	28.0	16.0
*Academic level*										
No study	1.2	1.6	0.0	7.7	4.2	0.0	0.0	0.0	0.0	0.0
Primary school	24.3	25.2	24.4	15.4	25.0	37.1	0.0	0.0	0.0	0.0
Secondary school	47.9	47.2	44.4 *^a^	46.1	58.3	34.3 *^b^	85.7 *^a,b^	100	0.0	100
High school	21.4	21.3	26.8	30.8	8.3	22.9	14.3	0.0	0.0	0.0
College	5.2	4.7	4.4	0.0	4.2	5.7	0.0	0.0	100	0.0
*Social class*										
Housewife	80.9	81.9	80.0	76.9	91.6	80.0	85.7	100	0.0	100
Student	1.2	3.1	4.4	0.0	0.0	2.8	14.3	0.0	0.0	0.0
Employed	8.7	6.3	11.1	0.0	0.0	8.5	0.0	0.0	0.	0.0
Day worker	0.6	0.8	0.0	7.7	0.0	0.0	0.0	0.0	0.0	0.0
Dealer	4.6	4.7	2.2	7.7	4.2	5.7	0.0	0.0	100	0.0
NA	4.0	3.1	2.2	7.7	4.2	2.8	0.0	0.0	0.0	0.0
*Civil status*										
Single	13.3	19.5	17.8	7.7	12.5	25.7	28.6	0.0	100	100
Married	20.3	14.8	22.2	15.4	8.3	8.6	28.6	0.0	0.0	0.0
Cohabitant	66.3	63.3	60.0	69.2	70.8	65.7	42.8	100	0.0	0.0
Widow	0.0	2.3 *	0.0 *^c^	7.7	8.3 *^c^	0.0	0.0	0.0	0.0	0.0
*Previous pregnancy*										
0	27.9	39.1	40.0	23.1	50.0	31.4	57.1	0.0	0.0	100
1	27.3	21.9	13.3	23.1	8.3	42.9	14.3	100	0.0	0.0
2	19.2	17.9	22.2	23.1	12.5	14.3	28.6	0.0	0.0	0.0
>2	25.6	21.1	24.4	30.7	29.2	11.4	0.0	0.0	100	0.0
*Self-reported use*										
Illicit drugs	0.0	6.3	0.0	0.0	8.3	14.3	0.0	0.0	0.0	0.0
Tobacco	1.7	10.9 *^d^	6.7	0.0	16.7	20.0	0.0	0.0	0.0	0.0
Alcohol	11.6	13.3	8.9	15.4	8.3	22.8	14.3	0.0	0.0	0.0
Inhalant	0.0	2.3	6.7	0.0	0.0	0.0	0.0	0.0	0.0	0.0
*Offspring’s pathologies*										
YES	18.5	12.6	13.3	15.4	16.7	8.6	14.3	0.0	0.0	0.0

NA: not available; 6-MAM: 6-Monoacetylmorphine; N,N-DMT: N-N-dimethyltryptamine; * *p* < 0.05: statistically significant; *^a^
*p* < 0.05: statistically significant differences between cannabis and NPSs groups; *^b^
*p* < 0.05: statistically significant differences between polydrug and NPSs groups; *^c^
*p* < 0.05: statistically significant differences between cannabis and methamphetamine groups; *^d^
*p*: statistically significant differences between tobacco self-reported use in positive and negative illicit drugs group.

**Table 5 pharmaceuticals-15-00382-t005:** Sociodemographic profiles associated with gestational exposure to tobacco smoke assessed by hair testing.

Variables (%)	Maternal Hair Negative to Nicotine (*n* = 217)	Maternal Hair Positive to Nicotine (*n* = 83)
*Nationality*		
Mexican	99.1	97.6
Others	0.4	2.4
NA	0.4	0.0
Age (mean ± SD)	24.6 ± 6.6	23.0 ± 5.2 *
*Academic level*		
No study	1.4	1.3
Primary school	19.4	38.5 *
Secondary school	52.1 *	36.1
High school	21.2	21.7
College	5.9 *	2.4
*Social class*		
Housewife	82.0	79.6
Student	1.8	2.4
Employed	8.8	4.8
Day worker	0.5	1.2
Dealer	4.6	4.8
NA	2.3	7.2
*Civil status*		
* Single*	17.1	13.3
* Married*	21.7 *	8.4
* Cohabitant*	60.8	75.9 *
* Widow*	0.4	2.4
*Previous pregnancy*		
0	35.0	26.5
1	21.7	33.7
2	20.3	14.5
>2	23.0	25.3
*Self-reported use*		
Illicit drugs	0.5	7.2
Tobacco	2.8	13.3 *
Alcohol	12.4	12.0
Inhalant	1.4	0.0
*Self-reported tobacco exposure*		
YES	37.2	57.8 *
*Self-reported exposure to familiar ETS*		
Tobacco	33.2	61.4 *
*Offspring’s pathologies*		
YES	17.1	13.3

NA: not available. * *p* < 0.05: statistically significant.

**Table 6 pharmaceuticals-15-00382-t006:** Newborn profiles associated with negative and positive illicit drugs and tobacco biomarkers detected in women hair.

Maternal Hair										
Variables (%)	All Cases (*n* = 300)	Negative for Illicit Drugs (*n* = 173)	Positive for Illicit Drugs (*n* = 127)	Negative for Nicotine (*n* = 217)	Positive for Nicotine (*n* = 83)	Positive Only for Cannabis (*n* = 45)	Positive Only for Cocaine (*n* = 13)	Positive Only for Methamphetamine (*n* = 24)	Positive than One Drug (*n* = 35)	Positive for NPSs (*n* = 7)
** *Newborn characteristics* **										
Weight (g, mean ± SD)	3004.6 ± 560.7	2969.1 ± 582.8	3053.0 ± 527.6	3005.6 ± 602.0	3001.9 ± 437.9	3187.8 ± 633.5 *	3025.8 ± 591.8	2911.7 ± 396.7	3019.7 ± 427.3	2844.3 ± 439.5
Height (cm, mean ± SD)	49.2 ± 3.3	49.0 ± 3.6	49.4 ± 2.7	49.1 ± 3.6	49.4 ± 2.2	50.1 ± 3.5	49.5 ± 3.1	49.1 ± 1.6	49.0 ± 1.9	48.9 ± 2.7
Gestational age (week, mean ± SD)	38.5 ± 2.3	38.4 ± 2.6	38.6 ± 2.0	38.5 ± 2.6	38.5 ± 1.6	38.5 ± 2.6	38.5 ± 2.1	38.9 ± 1.5	38.6 ± 1.4	39.4 ± 1.6
Head circumference (cm, mean ± SD)	33.9 ± 2.0	33.9 ± 2.2	34.0 ± 1.7	34.0 2.2	33.9 ± 1.3	34.3 ± 1.8	33.7 ± 2.1	33.9 ± 1.3	33.9 ± 1.4	33.0 ± 2.4
** *Offspring’s pathologies* **										
*One pathology*										
no pathology	84.0	81.5	87.4	82.9	86.7	86.7	84.6	83.3	91.4	85.7
Premature	2.3	3.5	0.8	2.8	1.2	0.0	0.0	0.0	2.8	0.0
SDR ^a^	1.0	0.0	2.4	0.9	1.2	2.2	0.0	8.3	0.0	0.0
TTRN ^b^	1.7	2.9	0.0	2.3	0.0	0.0	0.0	0.0	0.0	0.0
Neonatal sepsis	0.7	1.2	0.0	0.0	2.4	0.0	0.	0.0	0.0	0.0
Metabolic disturbances	2.0	2.3	1.6	2.8	0.0	2.2	0.0	4.2	0.0	0.0
*More one pathology*										
Premature and SDR	3.3	4.6	1.6	3.2	3.6	2.2	0.0	4.2	0.0	0.0
Premature and TTRN	1.6	1.2	2.4	0.9	3.6	0.0	7.7	0.0	2.8	14.3
Premature and neonatal sepsis	0.7	0.0	1.6	0.5	1.2	0.0	7.7	0.0	2.8	0.0
Premature and metabolic disturbances	0.3	0.0	0.8	0.5	0.0	2.2	0.0	0.0	0.0	0.0
Premature, SDR and neonatal sepsis	1.7	2.3	0.8	2.3	0.0	2.2	0.0	0.0	0.0	0.0
Premature, TTRN and neonatal sepsis	0.3	0.0	0.8	0.5	0.0	2.2	0.0	0.0	0.0	0.0
TTRN and neonatal sepsis	0.3	0.6	0.0	0.5	0.0	0.0	0.0	0.0	0.0	0.0

^a^ SDR: neonatal respiratory distress syndrome; ^b^ TTRN: transient tachypnea of the newborn; * *p* < 0.05: statistically significant difference between the cannabis group and the group with hair negative for illicit drugs.

## Data Availability

All data generated or analyzed during this study are included in this published article.

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
