# Peer review of "Prevalence of Licit and Illicit Drugs Use during Pregnancy in Mexican Women"

_pharmaceuticals, 2022, doi:10.3390/ph15030382_

Round 1
Reviewer 1 Report
This manuscript “Prevalence of Licit and Illicit Drugs Use During Pregnancy in Mexican Women” investigates influence of licit and illicit drugs on fetal development and pregnancy in Mexico. This is not something that is often considered in developed countries, but in developing countries like Mexico, it is an important problem that needs to be solved, and this study is expected to help. This study is expected to contribute to the solution of this problem. Since licit and illicit drugs were measured by LC-MS, the quantitative values of the drugs should be included in this study. To begin with, please clarify at what drug concentration the presence or absence of a drug is determined. Then, base your discussion on the various drug concentrations.
Author Response
Reviewer 1
This manuscript “Prevalence of Licit and Illicit Drugs Use During Pregnancy in Mexican Women” investigates influence of licit and illicit drugs on fetal development and pregnancy in Mexico. This is not something that is often considered in developed countries, but in developing countries like Mexico, it is an important problem that needs to be solved, and this study is expected to help. This study is expected to contribute to the solution of this problem. Since licit and illicit drugs were measured by LC-MS, the quantitative values of the drugs should be included in this study. To begin with, please clarify at what drug concentration the presence or absence of a drug is determined. Then, base your discussion on the various drug concentrations.
In agreement with reviewer’ comment, we added a comment at lines 219-224: “The concentration of the detected psychoactive drugs would have been important to differentiate between sporadic and chronic consumption during gestation. Unfortunately, our develop method focused only in the identification of most possible pharmacologically active substances, even in the absence of pure standards so that quantitative analyses were not performed. In any case, the mere presence of a drug in the hair of the pregnant women was an index of consumption, in any case harmful and prohibited during the development of the fetus”
Reviewer 2 Report
Interesting study of the use of illicit and licit drugs during pregnancy in an Mexican cohort using cm.
The objective of the study is clear, the used methods and statistical tools are adequate, the obtained results were thoroughly discussed.
One major comment: it is a pity that an alcohol consumption study was not performed using EtG in hair determination as the information of alcohol consumption is available.
Here are my minor comments:
line 36-37: the sentence "use of..." should be rephrased as it is not clear what is really meant.
line 232 replace "but" by "with"
Line 238 correct "affectoffspring" to "affect offspring"
In the tables "NR" should be replaced by "NA".
Author Response
Reviewer 2
Interesting study of the use of illicit and licit drugs during pregnancy in an Mexican cohort using cm.
The objective of the study is clear, the used methods and statistical tools are adequate, the obtained results were thoroughly discussed.
One major comment: it is a pity that an alcohol consumption study was not performed using EtG in hair determination as the information of alcohol consumption is available.
In agreement with reviewer’ comment, we addeda comment at lines 224-227: “ Another limitation of the present study was that developed method showed low sensitivity for the measurement of gestational alcohol consumption biomarker, hair ethylglucuronide [27]. As reported, a specific sample extraction coupled to an exclusive UHPLC-HRMS method for EtG is currently under evaluation”.
Here are my minor comments:
line 36-37: the sentence "use of..." should be rephrased as it is not clear what is really meant.
In agreement with the reviewer’s comment, we rephrased the sentence in the abstract
line 232 replace "but" by "with"
AMENDED.
Line 238 correct "affectoffspring" to "affect offspring"
AMENDED.
In the tables "NR" should be replaced by "NA".
AMENDED.
Round 2
Reviewer 1 Report
I do not understand the authors' answer. Even if the authors don't quantify it, how do the authors determine if it is there or not? For each drug, the authors were probably judging that it was "present" when it was above a certain concentration. The authors need to specify the specific values.
Author Response
I do not understand the authors' answer. Even if the authors don't quantify it, how do the authors determine if it is there or not? For each drug, the authors were probably judging that it was "present" when it was above a certain concentration. The authors need to specify the specific values.
In agreement with reviewer comment, we added a sentence in red at line 277:
Since proposed methodology was only intended for qualitative screening of hair samples limits detection and of identification were estimated as previously reported [27].Hair samples were considered positive when substances were found above limit of identifications ranging from 0.02 to 0.10 ng/mg hair.